# ADDITION CIRCUIT: HOW LLMS ADD IN THEIR HEADS USING STATE VECTORS

## ABSTRACT

Large Language Models (LLMs) are often treated as black boxes, yet many of their behaviours suggest the presence of internal, algorithm-like structures. We present *addition circuit* as a concrete, mechanistic example of such a structure: a sparse set of attention heads that perform integer addition. Focusing on two popular open-source models (Llama 3.1 8B and Llama 3.1-70B), we make the following contributions. (i) We extend prior work on two-argument addition to the multi-argument setting, showing that both models employ fixed subsets of attention heads specialized in encoding summands at specific positions in addition prompts. (ii) We introduce *state vectors* that efficiently capture how models represent summands in their activation spaces. We find that each model learns a common representation of integers that generalizes across prompt formats and across six languages, whether numbers are expressed as Arabic digits or word numerals.

## 1 INTRODUCTION

Large language models (LLMs) have demonstrated impressive capabilities in symbolic tasks, including arithmetic, logic, and algorithmic reasoning, despite being trained on natural language corpora with no explicit supervision for such skills. However, the exact mechanism by which these computations are performed remains poorly understood. A deep understanding of their mathematical reasoning capabilities becomes increasingly important as LLMs become more and more common components of larger systems, often supporting human decision makers in critical domains such as finance, healthcare and legal.

To address this gap, we study how Llama 3.1 8B and Llama 3.1 70B perform addition, extending prior work beyond the standard two-argument, single-token setting (Zhang et al., 2024; Zhou et al., 2024). Specifically, we analyze multi-argument addition of up to five integers where each summand is a single token. Our analysis reveals a sparse set of heads reliably assigning high attention scores to the arguments of addition, reusable *addition circuit* that generalizes across prompt templates and number formats (Arabic digits and word numerals) in six languages.

Additionally, we introduce *state vectors* which accurately capture the representation of each summand. State vectors let us localize which parts of the model encode each argument and enable us to perform targeted causal interventions to test whether those representations are necessary and sufficient for correct addition.

## 2 RELATED WORK

**Mechanistic Interpretability** Mechanistic Interpretability aims at explaining model's computations. Olah et al. (2020) argued that it is possible to discover *circuits* - small subsets of model's parameters responsible for accomplishing given task. Popular examples of circuits include modular addition circuit (Nanda et al., 2023) and the Indirect Object Identification circuit (Wang et al., 2022) (Ameisen et al., 2025) is a recent broad study on circuit discovery

**Interventions and Vector Arithmetic** Multiple works explored the idea of localizing specific knowledge or computation in LLM's weights by modifying model's activations during inference and measuring the change in its downstream behaviour (logits difference, task accuracy). Activation

patching (Zhang & Nanda, 2024; Goldowsky-Dill et al., 2023; Turner et al., 2024), steering vectors (Panickssery et al., 2024; Arditi et al., 2024) and function vectors (Todd et al., 2024).

**LLMs and Arithmetic tasks** Recent studies have increasingly focused on understanding how large language models (LLMs) represent numbers and perform addition. Stolfo et al. (2023) investigates arithmetic circuits and identifies key *layers* transferring the information required for arithmetic tasks to the final token positions. Nanda et al. (2023) reverse-engineers the *a+b mod p* algorithm in a toy model, providing a weight-level interpretation of the learned mechanism. Zhou et al. (2024) demonstrates that GPT-2-XL leverages Fourier Features to perform addition. Similarly, Zhang et al. (2024) describes the circuits underlying arithmetic reasoning in Llama 2-7B, Llama 2-13B, and Mistral-7B, identifying a small set of attention heads that specialize in attending to operands and operators across a variety of arithmetic tasks, with their importance confirmed through attention knockout experiments. Concurrent to our work, Hu et al. (2025) examines the two-argument addition case and the number representations in Llama 3.1-8B when one of the summands is not given directly but has to be inferred in-context from a set of examples. Kantamneni & Tegmark (2025) study two-argument addition in GPT-J, Pythia-6.9B, and Llama 3.1-8B, showing that the models embed numbers along a multidimensional helix that can be described using just seven parameters, and apply a "clock" algorithm to compute sums. Finally, Cheng et al. (2025) proposes that LLMs solve arithmetic tasks in two stages: first by abstracting the problem, and then by performing the computation. The *addition circuit* introduced in our work corresponds to this second stage.

## 3 METHOD

### 3.1 BACKGROUND AND NOTATION

We focus on autoregressive, transformer-based language models Vaswani et al. (2017). Given a sequence of tokens $t_1, t_2, \ldots, t_n$ from a vocabulary $\mathcal{V}$, the model computes a probability distribution over the next token. Each token is first embedded into a $d_{\text{model}}$-dimensional space: $z_i^0 = \text{Embed}(t_i)$. The resulting vectors are then passed through the $L$ transformer layers. Each layer consists of a Multi-Head Attention (MHA) layer followed by a feedforward MLP layer[1]. At each layer, all hidden states are processed by a single MHA and a single MLP:

$$\text{att}_i^l = \text{MHA}^l(z_0^l, \ldots, z_i^l)$$
$$\text{mlp}_i^l = \text{MLP}^l(z_i^l + \text{att}_i^l)$$
$$z_i^{l+1} = z_i^l + \text{att}_i^l + \text{mlp}_i^l,$$

The sequence of stacked hidden states $\{z_{1:n}^0, z_{1:n}^1, \ldots, z_{1:n}^L\}$ is called the *residual stream* (Elhage et al., 2021). For readability, we will omit the layer index where it is clear from the context.

We adopt the notation consistent with Todd et al. (2024): the $k$-th attention head $\text{h}^k$ is parametrized with four matrices $W_Q{}^k, W_K{}^k, W_V{}^k \in \mathbb{R}^{d_{\text{model}} \times d_{\text{head}}}, W_O{}^k \in \mathbb{R}^{d_{\text{head}} \times d_{\text{model}}}$. The output of a given head $h$ is defined as:

$$\text{h}(z_{1:n}) = AVW_O, \quad \text{h}(z_{1:n}) \in \mathbb{R}^{n \times d_{model}},$$

where

$$Q = z_{1:n}W_Q, \quad Q \in \mathbb{R}^{n \times d_{\text{head}}}$$
$$K = z_{1:n}W_K, \quad K \in \mathbb{R}^{n \times d_{\text{head}}}$$
$$V = z_{1:n}W_V, \quad V \in \mathbb{R}^{n \times d_{\text{head}}}$$
$$A = \text{softmax}\left(\frac{QK^\top}{\sqrt{d_{\text{head}}}}\right), \quad A \in \mathbb{R}^{n \times n}$$

In the calculation of the attention-weight matrix $A$, all strictly upper-triangular entries ($i > j$) of $QK^\top$ are masked (set to $-\infty$) to prevent the earlier tokens from attending to the subsequent ones. The final output of the MHA can be calculated as the sum of the outputs of each of its heads $\text{att}(z_{1:n}) = \sum_{k=1}^{n_{\text{heads}}} h^k(z_{1:n})$, where $n_{\text{heads}} = \frac{d_{\text{model}}}{d_{\text{head}}}$ is the total number of attention heads per layer.

---

[1]Following Stolfo et al. (2023), we omit the Normalization Layers (Ba et al., 2016) and Positional Embeddings for brevity, as they are not central to our analysis.

This is mathematically equivalent to the original formulation in Vaswani et al. (2017) using the concatenation of all head outputs and then multiplying them by a single $\boldsymbol{W}_O$ matrix. The per-head notation used in this work makes it easier to study the effect of each attention head on the residual stream.

Additionally, models from the Llama family Grattafiori et al. (2024) implement self-attention using Group Query Attention (GQA). All heads belonging to the same group share $\boldsymbol{W}_K$ and $\boldsymbol{W}_V$ matrices for keys and values computation. The MLP layer is a gated MLP defined as $\mathrm{MLP}(\boldsymbol{z}_i) = \sigma(\boldsymbol{z}_i \boldsymbol{W}_{\mathrm{gate}}) \circ (\boldsymbol{z}_i \boldsymbol{W}_{\mathrm{up}}) \boldsymbol{W}_{\mathrm{down}}$, where $\boldsymbol{W}_{\mathrm{gate}}, \boldsymbol{W}_{\mathrm{up}} \in \mathbb{R}^{d \times 4d}$, $\boldsymbol{W}_{\mathrm{down}} \in \mathbb{R}^{4d \times d}$, $\sigma$ is a SiLU activation function (Elfwing et al., 2017), and $\circ$ represents the element-wise multiplication (Liu et al., 2021).

Finally, the hidden state $\boldsymbol{z}_n^L$ corresponding to the last token position in the output of the final layer is multiplied by the unembedding matrix $\boldsymbol{U} \in \mathbb{R}^{d \times |\mathcal{V}|}$ and the softmax function is applied to the resulting vector to obtain probability distribution over the model's vocabulary $\mathcal{V}$. In this analysis, we focus on generating the single next token by selecting the one with the highest probability.

## 3.2 Datasets

We construct a dataset of addition prompts with up to 5 arguments. We observed that the simple prompt $x_1 + \cdots + x_N =$ often results in the model answering by repeating the first argument of addition. The following prompt templates were chosen to avoid this behaviour and incentivize the model to directly output the result of addition without [2]:

$$\text{`Hence, } x_1 + x_2 + [\ldots] + x_N \text{ is equal to '} \qquad \text{(Prompt Format 1)}$$

and

$$\text{`Alice has } x_1 \text{ apples, Bob has } x_2 \text{ apples, } [\ldots] \text{ has } x_N \text{ apples. In total, they have '}$$
$$\text{(Prompt Format 2)}$$

We vary the number of arguments $N \in \{2, 3, 4, 5\}$. In each prompt replace the argument placeholders $x_i$ with either Arabic digits or numerals in one of the following languages: English, Spanish, French, German, Portuguese, Italian. We only consider values $x_i \in [0, 99]$ to make sure the result can be also represented by a single token. The exact translations of prompts can be found in Appendix A

## 3.3 Models

We test two transformer-based models: Llama 3.1 8B and Llama 3.1 70B. All three use Grouped-Query-Attention mechanism. The configuration parameters of each of the models are presented in Table 1. Both models tokenize all integers in the range $[0, \ldots, 999]$ as a single token.

Table 1: Comparison of model architectures.

| Model | Layers | Heads / Layer | Residual Stream Dimension | Number of Heads per Group |
|-------|--------|---------------|---------------------------|---------------------------|
| Llama 3.1 8B | 32 | 32 | 4096 | 8 |
| Llama 3.1 70B | 80 | 64 | 8192 | 8 |

## 3.4 State Vectors

Let $H^\ell$ denote the set of all heads in layer $\ell$. For a given prompt $p$ and any set of heads $G \subseteq H^\ell$ let $v_{i,j}(p, G)$ be the head-restricted contribution from the hidden state at position $i$ to the hidden state at position $j$:

$$v_{i,j}(p, G) \;=\; \sum_{h \in G} \boldsymbol{A}_{j,i}^{\ell,h}(p) \, \boldsymbol{z}_i^{\ell-1}(p) \, \boldsymbol{W}_V^{\ell,h} \boldsymbol{W}_O^{\ell,h}, \qquad (1)$$

---

[2]The code used to construct the dataset and perform all the experiments is available at https://anonymous.4open.science/r/addition-6CF9/

where $\boldsymbol{A}_{j,i}^{\ell,h}$ denotes the attention score in $j$-th row and $i$-th column of the attention matrix $\boldsymbol{A}^{\ell,h}$. We define *state vector* $s_{i,j}(\mathbb{P}, G)$ as the average head-restricted contribution $v_{i,j}(p, G)$ over a set of prompts $\mathbb{P}$:

$$s_{i,j}(\mathbb{P}, G) = \frac{1}{|\mathbb{P}|} \sum_{p \in \mathbb{P}} v_{i,j}(p, G) \tag{2}$$

By selecting an appropriate pair of indices and an appropriate set of prompts and heads, we can interpret *state vectors* as abstract representations of concepts. In section 4, we demonstrate how both Llama 3.1 8B and Llama 3.1 70B represent the arguments of addition as *state vectors* and how they reuse this representation across diverse prompts requiring integer addition.

# 4 ADDITION CIRCUIT

## 4.1 FINDING THE ADDITION CIRCUIT: MULTIPLE SINGLE-TOKEN ARGUMENTS

We will show that the *addition circuit* encodes each argument of addition $x_i$ using small sets of heads $G_{x_i}^\ell$ located in the middle layers of the model.

To better illustrate this idea, for each number of arguments $2 \leq N \leq 5$ and argument position $1 \leq i \leq N$ we sampled 100 prompts and compute the average attention scores for all heads in all layers across all prompts. We do this separately for both Prompt Format 1 and Prompt Format 2. In Figure 1 we present the average attention scores for layer 15. As can be seen in the plot, on average, head L15H13 specializes in attending to the argument $x_2$. Similarly, head L15H3 specializes in attending to $x_3$. Additional figures for other layers and Llama 3.1 70B can be found in Appendix B.

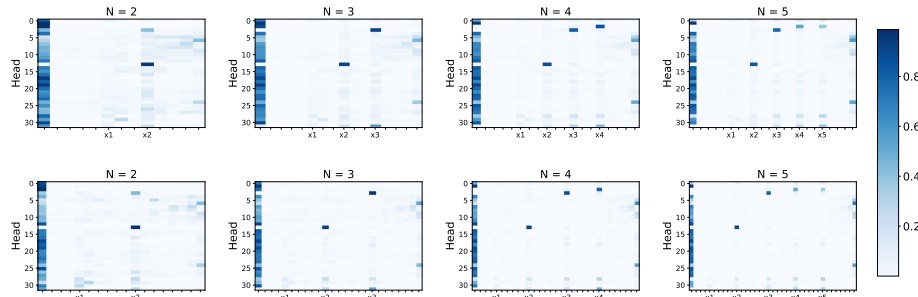

Figure 1: Average attention patterns for Llama 3.1 8B Layer 15, across 100 prompts for different numbers of arguments $N$ using Prompt Format 1 in the top row and Prompt Format 2 in the bottom row.

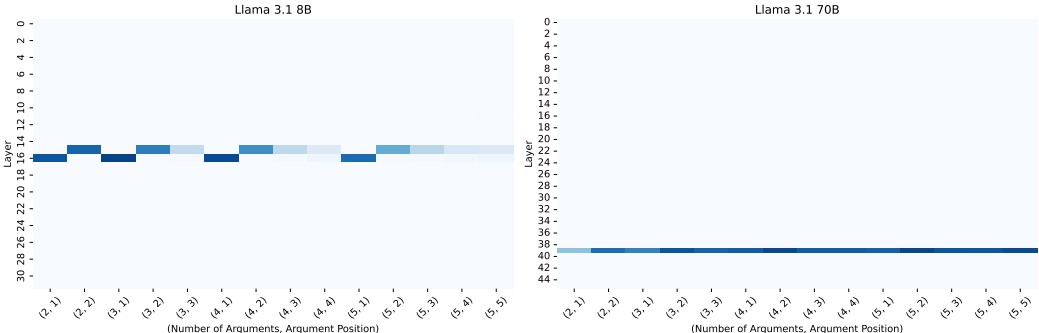

Figure 2: Intervention accuracy on Llama 3.1 8B and Llama 3.1 70B models by intervening on a single layer for different total number $N$ of arguments and argument positions

To formalize this idea, we begin by finding a subset of layers responsible for processing the arguments of addition. Let $\mathbb{P}_{N,i,c}$ be a set of 100 randomly sampled N-argument prompts following Prompt Format 1 such that all arguments are integers in the range $[0, \ldots, 99]$, and for each prompt $x_i = c$. Since all arguments are represented as a single token, all prompts in $\mathbb{P}_{N,i,c}$ have the same length $|p|$. For all $2 \leq N \leq 5, 1 \leq i \leq N, 0 \leq c \leq 99$, we compute a state vector $s^{\ell}_{\text{pos}(x_i),|p|}(\mathbb{P}_{N,i,c}, H^{\ell})$. This *state vector* captures the average contribution from the hidden state representing $x_i = c$ to the last token position at layer $\ell$.

We use the *state vectors* computed above to perform the intervention. We sample a random $N$-argument prompt $p$ with $x_i = c$ and use it to perform a single forward pass through the model. Next we randomly sample a new value $c' \neq c$ and we intervene on the model's computations by modifying the hidden state in a single layer $\ell$ at the last token position in the residual stream directly after the multi-head attention module:

$$z^{\ell}_{|p|} \leftarrow z^{\ell}_{|p|} - s^{\ell}_{\text{pos}(x_i),|p|}(\mathbb{P}_{N,i,c}, H^{\ell}) + s^{\ell}_{\text{pos}(x_i),|p|}(\mathbb{P}_{N,i,c'}, H^{\ell}) \tag{3}$$

Intuitively, this operation corresponds to erasing the information about the value $c$ of the $i$-th argument from the residual stream and replacing it with a new value $c'$. Assuming that the sum of the $N$ arguments in the original prompt was equal to $S$, we consider the intervention to be successful only if the first token returned by the model after the intervention is equal to $S - c + c'$.

We perform 100 times for every layer and for every number of arguments and argument position $i$ and report the results in Figure 2. Notably, the interventions have a high success rate only in layers 15, 16 for Llama 3.1 8B model and layer 39 for Llama 3.1 70B model, confirming that modifying the outputs of these layers has a causal effect on the result of addition.

Next, we want to narrow down a subset of heads specialized in encoding each of the arguments of addition. We define sets $G^{\ell}_{x_i}$ as $k$ heads with the highest average attention scores assigned to $x_i$ for 5-argument prompts. We set $k = 3$ for Llama 3.1 8B and $k = 6$ for Llama 3.1 70B. The exact heads are listed in the Appendix C. We test the *addition circuit* described above by performing 100 causal interventions as described above, this time using *state vectors* computed using subsets of heads $G^{\ell} \subset H^{\ell}$ instead of the full set of heads. We report the results in Figure 3

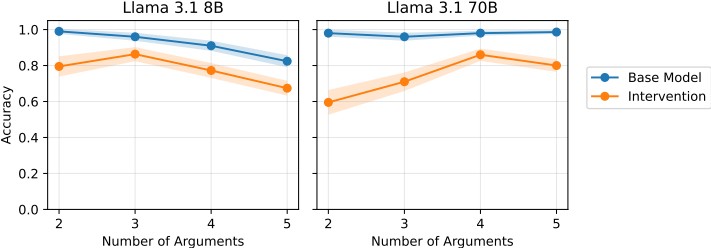

Figure 3: Base Model (no intervention) and Intervention accuracy using *state vectors* over 100 prompts in Prompt Format 1 An intervention is considered successful if the model's output token matches the sum of arguments with $x_i = c$ replaced by $x_i = c'$. Shaded regions indicate the 95% confidence interval.

Across both models and all argument positions, intervention accuracy remains above 60% and closely follows the baseline accuracy. This provides strong evidence that (i) the *state vectors* capture how the model represents each argument of addition in prompts following Prompt Format 1, and (ii) modifying the selected head subsets identified above aha a causal influence on the final sum.

## 4.2 ADDITION CIRCUIT AND STATE VECTORS GENERALIZE TO OTHER PROMPT FORMATS

So far, we demonstrated that *state vectors* can be used to intervene on activations of the model to causally change the value of a given argument when the input prompt follows the Prompt Format 1. In thise section, we demonstrate that the discovered *state vectors* generalize to other prompt formats.

We show this by applying **the same** *state vectors* computed in the previous section using Prompt Format 1 by applying them to prompts in Prompt Format 2.

In contrast to the previous prompt format, Prompt Format 2 does not include mathematical symbols other than integers, and the model needs to infer the need for adding numbers from the context. Additionally, prompts in this format contain nouns and verbs, which could potentially obscure the addition task and affect the values of hidden states in the residual stream.

We repeat the causal interventions as described in the previous paragraph; however, we do not compute new *state vectors* using this new prompt format, but instead, we directly apply the *state vectors* computed earlier using Prompt Format 1. We report the results in Figure 4

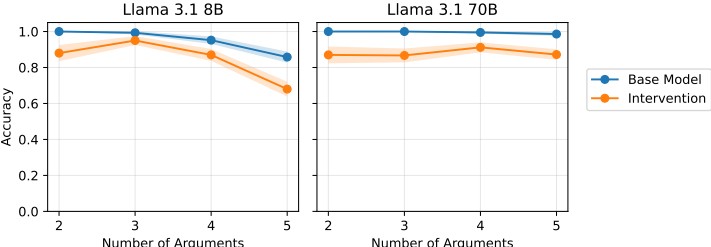

Figure 4: Base Model (no intervention) and Intervention accuracy using *state vectors* computed for Prompt Format 1, over 100 prompts in Prompt Format 2. An intervention is considered successful if the model's output token matches the sum of arguments with $x_i = c$ replaced by $x_i = c'$. Shaded regions indicate the 95% confidence interval.

As can be seen in the plot, the intervention accuracy closely follows the base accuracy for both models. This shows that *state vectors* are not tied to a specifc prompt template but generalize across prompts for which models have to perform addition.

### 4.3    STATE VECTORS ARE SEMANTIC ENTITIES

To assess the extent to which *state vectors* generalize, we perform interventions using the same *state vectors* on prompts in other languages. Instead of representing $x_i$ using Arabic digits, we use numerals in different languages, for example, "twenty-seven" instead of "27". This significantly changes how each of the prompts is processed, since numerals are represented with a variable number of tokens. Moreover, prompts can have a variable length, and numerals can start at different token positions depending on the exact values of previous arguments.

We intervene on the activations of the model presented with prompt Prompt Format 1, where the entire sentence and all arguments are translated to one of the 6 languages: English, German, Italian, Spanish, Portuguese and French. The exact translations are listed in the A. The results are summarized in Figure 5. The intervention accuracy closely follows the baseline accuracy of each of the models.

The fact that *state vectors* computed for Prompt Format 1 generalize well to a diverse set of other prompts shows that the model first identifies the location of each of the addition arguments and then uses the *addition circuit* to perform the calculation. It provides evidence for the existence of reusable, generalized representations emerging in the middle layers of each of the studied models, which can be approximated using *state vectors*.

### 4.4    ADDITION CIRCUIT: TWO MULTI-TOKEN ARGUMENTS

In this section, we briefly discuss how models add two arguments represented by multiple-tokens. As a motivational observation, consider an example prompt following the Prompt Format 1 with argument `123123123`, `45645645`. As illustrated in Figure 6 Llama 3.1 8B processes the summands in three digit groups from left to right, converting each group into a single token. Moreover, it can be seen from the plot that at each generated token, the model reuses the components from the addition circuit described above, namely head L16H21 attends to the currently processed token of

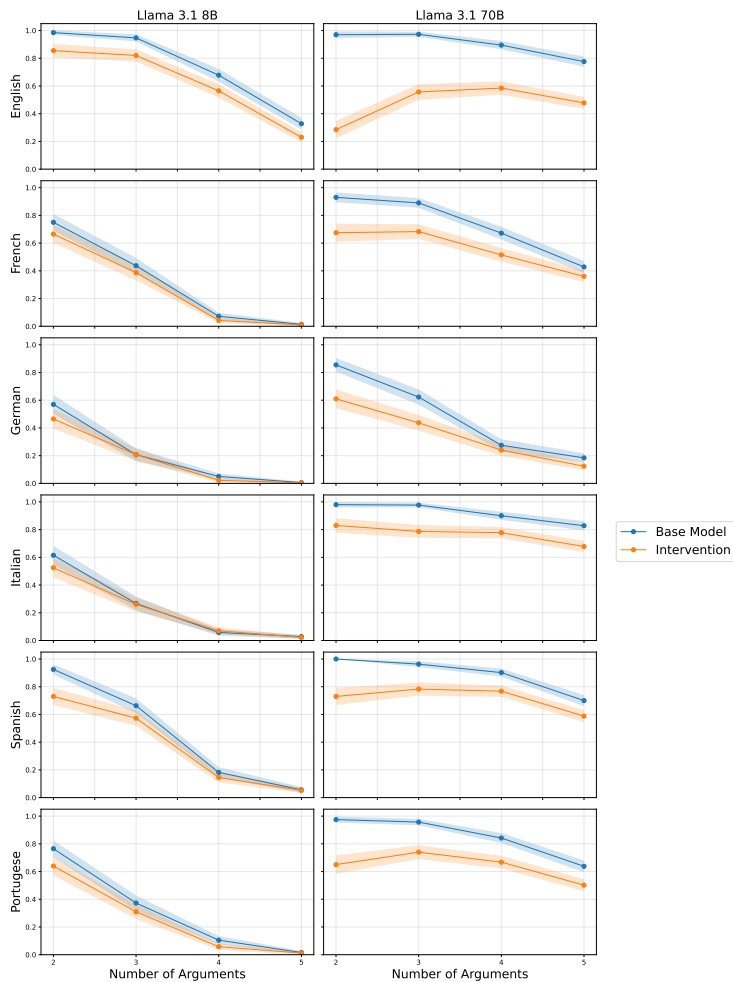

Figure 5: Intervention accuracy using *state vectors* computed for Prompt Format 1 with arabic numerals on Prompt Format 1 in natural languages. The intervention accuracy always closely follows the baseline accuracy of the model, demonstrating the causal effectiveness and the strong generalization of *state vectors*.

the first argument and head L15H13 attends to the currently processed token of the second argument. This suggests that the discovered *addition circuit* is a part of a larger circuit performing the addition of arguments represented with multiple tokens.

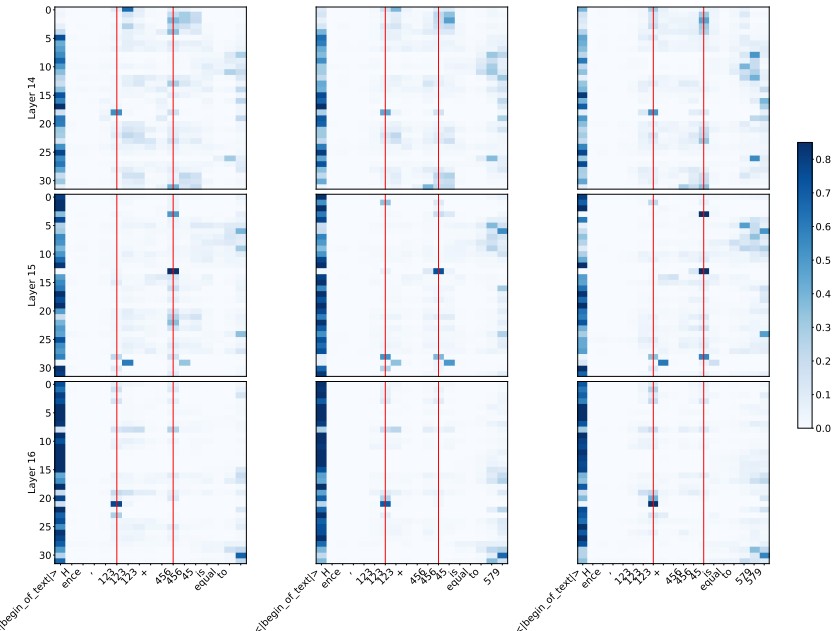

Figure 6: Attention patterns in Llama 3.1 8B for the addition of two integers represented as multiple tokens. Rows correspond to the attention heads in layers 14–16, columns show the attention patterns for three generated tokens. For each head, only the attention scores from the final token position are shown. The model processes both arguments by shifting its attention token by token from left to right - currently processed tokens are marked with red lines.

## 5  DISCUSSION

Our findings reveal a surprisingly modular and consistent mechanism by which the Llama 3.1 8B and Llama 3.1 70B models perform multi-argument arithmetic. Specifically, we show that individual attention heads consistently specialize in attending to specific argument positions in the addition tasks. This pattern is consistent across a wide variety of prompt styles and numeric representations, suggesting that the model has learned an abstract, position-aware representation of addition arguments.

One particularly striking observation is the independence of attention-based computations in the multi-argument, single-token setting. We demonstrate that the contribution of each argument to the final output is largely separable and can be manipulated independently using a small subset of attention heads. These subsets are stable across the prompt format or tokenization of the numbers indicating that the model learns representations of the summands which are semantic rather than symbolic.

This finding supports the hypothesis that LLMs learn canonical representations for integers. Our causal interventions confirm the causal role of these vectors: modifying the output of just a few attention heads can deterministically change the output of the model in a predictable and interpretable way.

This work contributes to the growing body of mechanistic interpretability research by providing tools and insights for locating and manipulating localized algorithmic behaviour within large-scale language models.

## LIMITATIONS

While our proposed framework is not limited to any particular architecture or domain, the experimental evidence provided in this paper focuses on LLama 3.1 family of models, and future work is needed to verify if these results translate to other architectures

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

## A  TRANSLATIONS

| Language | Translation of the prompt "Hence, 37 + 86 is equal to " |
| --- | --- |
| English | Hence, thirty seven + eighty six is equal to |
| German | Daher ist siebenunddreißig + sechsundachtzig gleich |
| Italian | Quindi, trentasette + ottantasei è uguale a |
| Spanish | Por lo tanto, treinta y siete + ochenta y seis es igual a |
| Portuguese | Portanto, trinta e sete + oitenta e seis é igual a |
| French | Ainsi, trente-sept + quatre-vingt-six est égal à |

# B    AVERAGE ACTIVATION PATTERNS

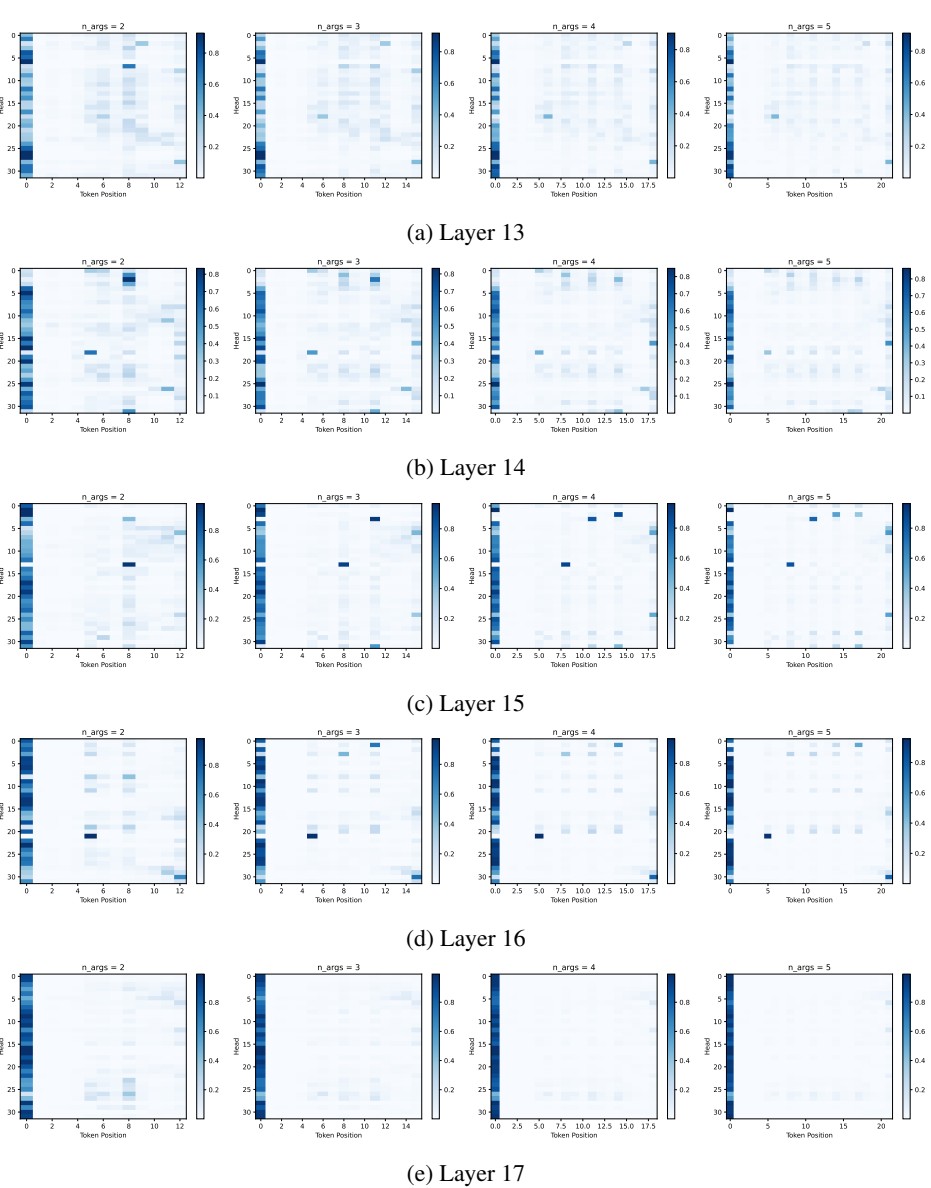

Figure 7: Average attention patterns over 100 random prompts for middle layers of the Llama 3.1 8B model for Prompt Format 1. Layers 14 and 15 are visibly more consistent at tracking the positions of addition arguments.

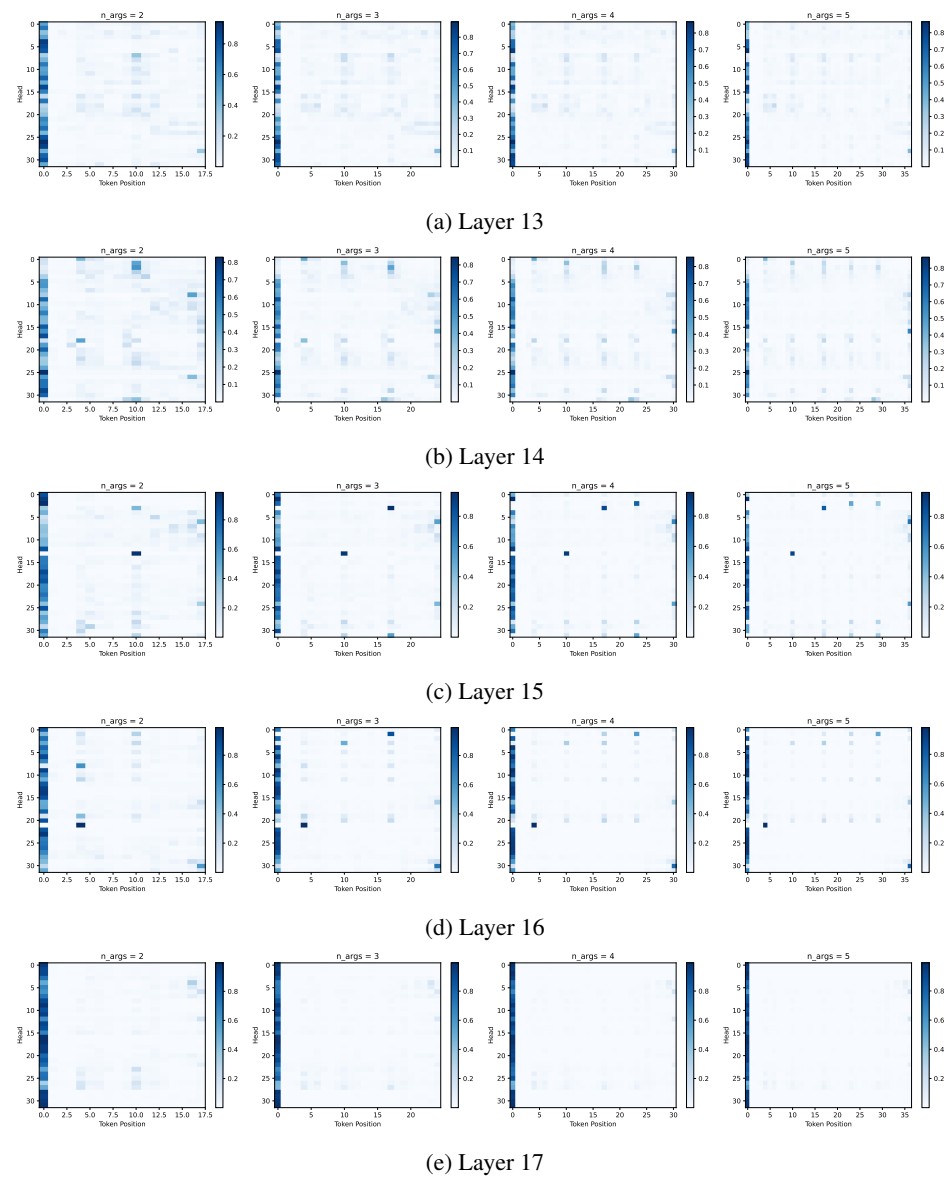

Figure 8: Average attention patterns over 100 random prompts for middle layers of the Llama 3.1 8B model for Prompt Format 2. Layers 14 and 15 are visibly more consistent at tracking the positions of addition arguments.

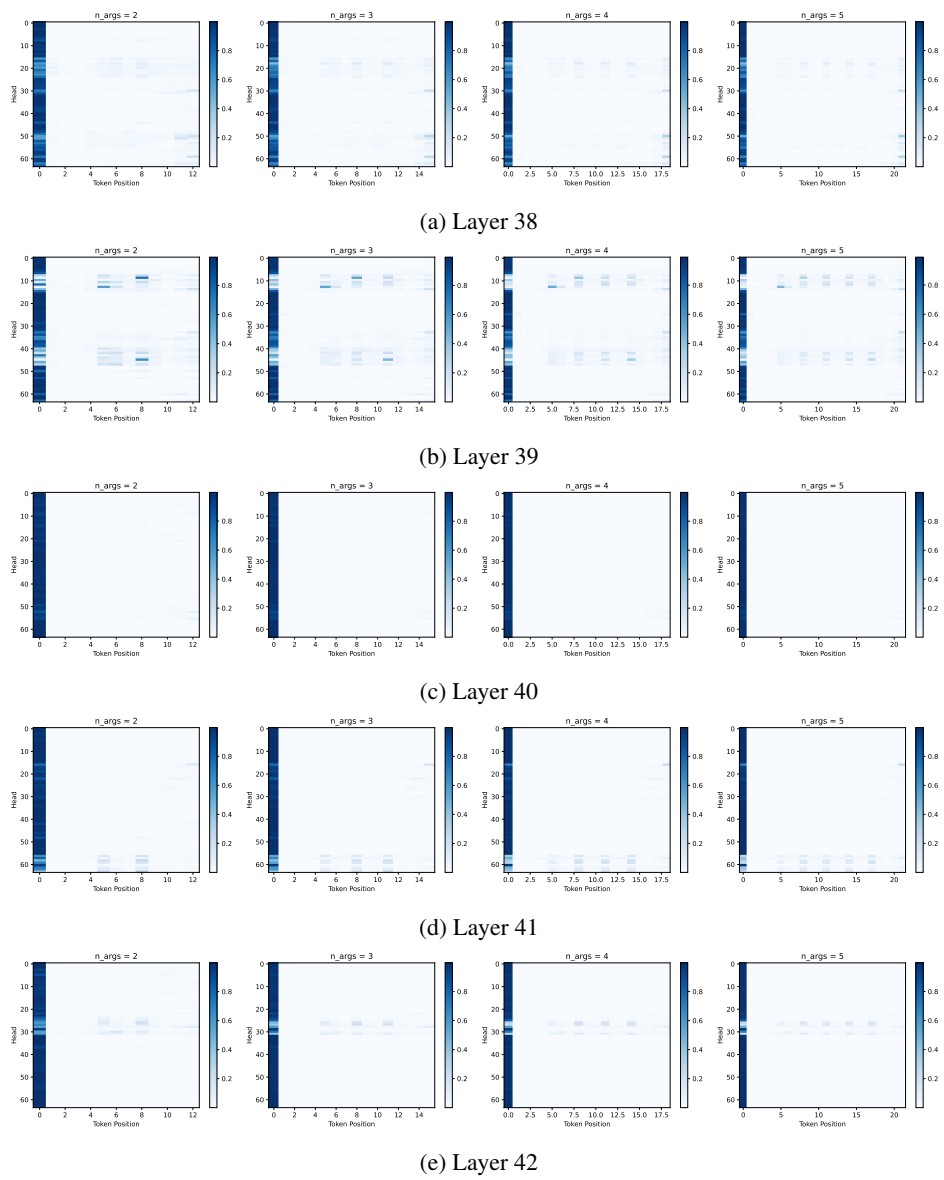

Figure 9: Average attention patterns over 100 random prompts for middle layers of the Llama 3.1 70B model for Prompt Format 1. Layer 39 is visibly more consistent at tracking the positions of addition arguments.

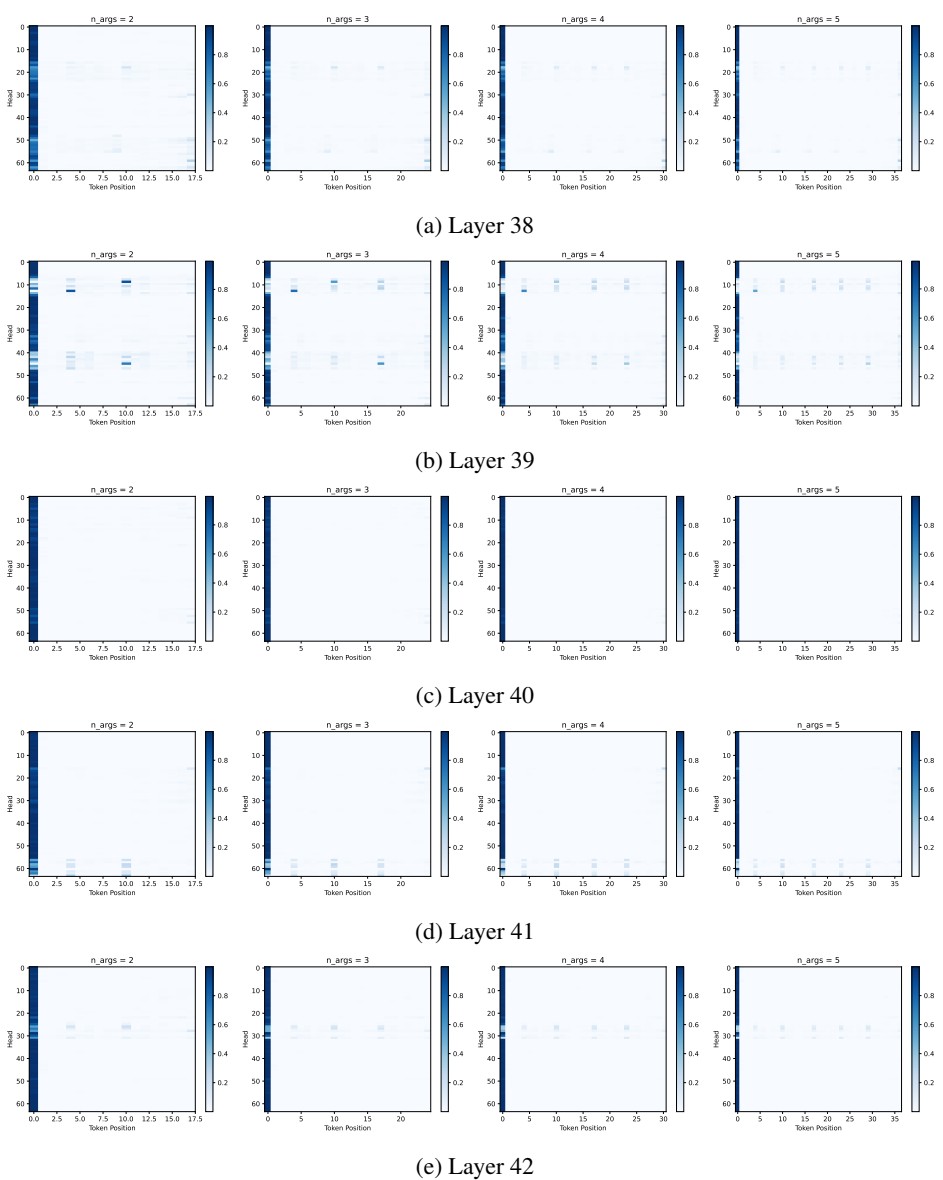

Figure 10: Average attention patterns over 100 random prompts for middle layers of the Llama 3.1 70B model for Prompt Format 2. Layer 39 is visibly more consistent at tracking the positions of addition arguments.

## C  APPENDIX

Table 2: Sets $G^{\ell}_{x_i}$ of attention heads encoding each of the addition arguments in the residual stream in multi-argument prompts for layers 15 and 16

| Model | Layer | $x_1$ | $x_2$ | $x_3$ | $x_4$ | $x_5$ |
|---|---|---|---|---|---|---|
| Llama 3.1 8B | 15 | H30 | H13 | H3 | H2 | H2 |
| | | H28 | H28 | H28 | H31 | H31 |
| | | H0 | H30 | H31 | H28 | H28 |
| | 16 | H21 | H3 | H3 | H1 | H1 |
| | | H19 | H20 | H20 | H20 | H20 |
| | | H20 | H19 | H19 | H3 | H11 |
| Llama 3.1 70B | 39 | H13 | H9 | H9 | H45 | H45 |
| | | H8 | H12 | H12 | H12 | H42 |
| | | H11 | H11 | H45 | H9 | H12 |
| | | H42 | H43 | H42 | H42 | H9 |
| | | H12 | H8 | H11 | H11 | H11 |
| | | H44 | H42 | H8 | H8 | H8 |
| | 41 | H59 | H63 | H63 | H63 | H59 |
| | | H56 | H59 | H59 | H59 | H56 |
| | | H63 | H56 | H58 | H58 | H63 |
| | | H58 | H58 | H56 | H56 | H58 |
| | | H62 | H62 | H62 | H62 | H62 |
| | | H61 | H61 | H61 | H61 | H61 |

