# OpenReview forum: "Addition Circuit: How LLMs Add in Their Heads using State Vectors"
_ICLR.cc/2026/Conference — Submitted to ICLR 2026_

### Official Review · Reviewer_Yo6t · 2025-10-15

**Soundness:** 1
**Presentation:** 2
**Contribution:** 1
**Rating:** 2
**Confidence:** 4

**Summary:**

This paper introduces the concept of an "addition circuit" in LLMs, demonstrating how Llama 3.1 models perform integer addition through specific attention heads. The authors extend prior work to multi-argument addition and introduce "state vectors" that capture how models represent summands across different prompt formats and languages. While the core findings are interesting, several claims appear overstated given the experimental evidence.

**Strengths:**

1. Novel extension to multi-argument addition: The work successfully extends beyond the standard two-argument setting to analyze addition with up to 5 arguments.
2. Cross-linguistic generalization: Testing across 6 languages (with both Arabic digits and word numerals) demonstrates interesting generalization properties.
3. State vector methodology: The introduction of state vectors provides a useful tool for understanding how models encode arguments in activation spaces.
4. Clear experimental design: The use of two distinct prompt formats helps test robustness of findings.

**Weaknesses:**

1. Methodological Issues
- The experimental setup is underexplained. No examples with two-digit arguments are shown, and it's unclear if test cases included "carry-one" calculations (e.g., 44+56=100).
- Without carry operations, the model may simply map digit pairs independently rather than implementing true addition.
The exception where simple prompts "x1 + ... + xN =" lead to repetition of the first argument is mentioned but not discussed.

2. Overstated Claims
- With ~60% intervention success for Prompt Format 1 and ~90% for Format 2, the conclusion that state vectors fully capture the addition mechanism is too strong.
- The variation in success rates suggests other undiscovered circuits may be at play, particularly for Format 1.
- The claim that state vectors are "abstract representations of concepts" needs qualification.

3. Technical Issues
- In Figures 1, 7 & 8, when N integers are added, N-1 attention heads are highly activated but the head for x1 is not obvious - this asymmetry needs explanation.
- Section 4.4 on multi-token arguments is the weakest and uses examples without carry operations, significantly simplifying the problem. Consider removing this section.

4. Presentation Issues
- Multiple typos: "aha a" → "have a", "thise" → "this", "specifc" → "specific"
- Figure 2's caption needs to be self-contained. The purpose of this figure is not clear from the caption alone.
- Font sizes in figures should be increased for readability
- Quotation marks formatting needs correction (use `` '' in LaTeX)

5. Meta comments / Recommendations
The paper makes interesting contributions but needs revision to:
- Clarify the experimental setup with explicit examples
- Moderate claims to match the evidence (60-90% success rates indicate partial circuit capture)
- Add analysis of failure cases
- Expand the related section to acknowledge more related papers (of which there are many).
- Section 3.1: GQA and SiLU are introduced too quickly without sufficient context
- Section 4.1: Please add specific test data examples in an appendix and reference here
- Section 4.3: The per-language variation in intervention success rates actually argues against a single universal circuit
- Section 5: The discussion should acknowledge that 60-90% intervention rates suggest partial rather than complete circuit identification
- Limitations: Understated given the evidence for additional circuits and untested complexity (carry operations)

6. Minor Points
- Figure 3 caption: "Format 1 An intervention" → "Format 1. An intervention"
- Better quote formatting throughout (`` '' in LaTeX)
- Consider dropping Section 4.4 as it's the least compelling

**Questions:**

1. What percentage of test cases required carry operations? This fundamentally affects what circuits we should expect.
2. Can you provide analysis of the failure cases where interventions didn't work? Is there a pattern (e.g., all involving carry operations)?
3. Why is there an asymmetry in attention patterns where x1 doesn't have an obvious dedicated head?
4. Have you considered that models might implement redundant circuits, as shown in prior work on circuit duplication?

---

### Official Review · Reviewer_cdHy · 2025-10-26

**Soundness:** 3
**Presentation:** 4
**Contribution:** 3
**Rating:** 6
**Confidence:** 4

**Summary:**

This paper investigates the internal mechanisms by which Large Language Models (LLMs), specifically Llama 3.1 8B and 70B, perform integer addition. Moving beyond prior work often focused on two-argument, single-token addition, the authors identify a sparse "addition circuit" composed of specific attention heads located in the middle layers of the models.

The core contributions are twofold:

1. **Multi-Argument Circuit Identification:** The paper demonstrates that distinct subsets of attention heads specialize in attending to summands based on their *position* within multi-argument addition prompts (up to 5 arguments).
2. **State Vectors and Generalization:** It introduces the concept of "state vectors," derived from the outputs of these specialized heads, which capture the model's internal representation of each summand. Crucially, the authors show these state vectors represent integers semantically, generalizing effectively across different prompt templates, across Arabic digits vs. word numerals, and across six different languages. The paper also briefly explores how these circuit components are reused for multi-token addition.


The mathematical definitions (head-restricted contribution, state vectors, intervention operator) are sound and internally consistent. The core experimental methodology, using causal interventions (activation patching via state vectors) to validate the function of identified attention heads, is appropriate and well-executed for demonstrating causal sufficiency. The results convincingly show that manipulating the derived state vectors for specific summands leads to the predicted change in the model's output sum with high accuracy, particularly in the single-token setting.

The paper is exceptionally well-written and clearly structured. The motivation is strong, the necessary background is provided concisely, and the core concepts like state vectors are introduced intuitively and defined formally. Figures effectively visualize the key findings, such as the positional specialization of attention heads (Fig. 1) and the success rates of interventions across various settings (Fig. 2-5). The narrative flows logically from circuit identification to causal validation and generalization testing.

The paper makes a solid contribution to mechanistic interpretability, particularly regarding arithmetic in LLMs. The primary contributions are:

1. Extending circuit analysis to **multi-argument** addition and identifying **position-specialized** heads.
2. Introducing **state vectors** as a conceptually clear and empirically effective tool for representing and causally manipulating numerical information within activations.
3. Providing compelling evidence for the **semantic generalization** of these internal representations across formats, numerals vs. words, and multiple **languages**.

While building on prior work identifying arithmetic mechanisms, these specific findings, especially the multi-argument aspect and the strong generalization results, represent a valuable advancement.

**Strengths:**

1. **Strong Causal Sufficiency:** The paper excels in providing rigorous evidence that the identified heads and state vectors are *sufficient* to causally control the model's addition output in the tested settings. The high success rate of interventions is compelling.
2. **Novelty in Multi-Argument & Positional Specialization:** The focus on >2 arguments and the discovery of position-specific roles for heads adds a layer of understanding beyond previous two-argument studies.
3. **State Vectors as an Interpretability Tool:** The state vector abstraction is clean, mathematically sound, and proven effective as a causal handle, potentially useful for future work.
4. **Impressive Generalization:** The demonstrated generalization across languages and formats is a standout result, strongly suggesting the model learns abstract number representations rather than surface-level correlations.
5. **Clarity and Reproducibility:** The methods and definitions are clear, aiding reproducibility. The consistency observed between 8B and 70B models is also encouraging.

**Weaknesses:**

1. **Lack of Necessity Proof:** While sufficiency is well-demonstrated, the paper doesn't rigorously establish *necessity*. It doesn't prove that *only* these heads/layers perform the crucial computation. Techniques like path patching ablations (zeroing out contributions from *only* these heads) or replacement controls (using random vectors) would be needed to strengthen the claim that this is *the* core mechanism.
2. **Head Selection Criterion:** Heads were selected based on high *attention scores*. This might miss heads with low attention but high causal impact (e.g., through the value pathway). Using causal attribution methods (like path patching) for head selection could provide a more functionally relevant circuit definition.
3. **Limited Scope (Single-Token Dominance):** The core quantitative results heavily rely on the single-token constraint $x_i \in [0, 99]$. The multi-token addition section is qualitative and illustrative (Fig. 6); generalizing the quantitative intervention analysis to multi-token numbers (including handling carries) would significantly broaden the impact.
4. **Intervention Locus Specificity:** Interventions are performed only at the last token position, immediately after the MHA block. It remains unclear if number information is significantly processed or altered by the MLP block or at earlier token positions within these layers.
5. **Interpretation of Generalization Accuracy:** When intervention accuracy "follows baseline" in challenging generalization settings (e.g., multilingual, Fig. 5), particularly where baseline accuracy drops, it shows compatibility but might overstate the degree of control if the model struggles with the task fundamentally. Reporting absolute accuracy post-intervention alongside baseline would be clearer.

This paper presents interesting findings on multi-argument addition circuits and introduces the potentially useful concept of state vectors, demonstrating strong causal sufficiency and impressive generalization. However, the claims regarding the identified circuit being *the* essential mechanism are not fully substantiated due to the lack of necessity experiments and potential limitations in head selection. Furthermore, the restriction to primarily single-token arithmetic limits the scope. While the work has clear merit, these limitations place it marginally above the threshold, though its rejection would not be strongly opposed.

**Questions:**

1. Could the authors elaborate on why attention scores were chosen for head selection over causal impact metrics (like path patching attribution), especially given that value-path contributions can be significant even with low attention?
2. Regarding necessity: Would the authors consider performing ablation experiments (e.g., zeroing the OV outputs of the identified heads $G^\ell_{x_i}$) to quantify the drop in addition accuracy, thereby assessing if this circuit is truly necessary?
3. How sensitive are the state vectors and intervention success rates to the specific set of prompts $P$ used for averaging in Eq. 2? Was robustness checked across different sampling distributions for $P$?
4. For the multi-token addition case (Sec 4.4), could the state vector approach be adapted, perhaps by defining state vectors for digit groups or carry operations, to allow for quantitative causal interventions similar to those in the single-token setting?

---

### Official Review · Reviewer_uWty · 2025-10-30

**Soundness:** 2
**Presentation:** 1
**Contribution:** 1
**Rating:** 2
**Confidence:** 3

**Summary:**

This paper investigates the internal mechanism of Llama 3.1 models in performing integer addition. The authors identify a small set of attention heads (``addition circuit’’) responsible for encoding summands and propose ``state vectors’’ to represent how models encode individual integers. They claim these state vectors generalize across prompt templates, number formats, and languages.

**Strengths:**

1.	Provides reproducible code and clear experiment descriptions.
2.	Presents consistent empirical observations linking attention heads to arithmetic behavior.

**Weaknesses:**

1.	The introduction is too brief and fails to articulate a clear motivation or position within existing research. It neither explains why multi-argument addition is important nor summarizes current progress in mechanistic interpretability.
2.	The paper cites a limited number of references, most of which are unpublished arXiv preprints, indicating insufficient engagement with peer-reviewed or foundational literature.
3.	The core idea replicates existing ``addition circuit’’ analyses, such as `Interpreting and Improving Large Language Models in Arithmetic Calculation’, extending them only from two to multiple operands without methodological innovation.
4.	Only two Llama 3.1 variants are tested, restricted to small integer ranges (0–99). No cross-architecture or multi-layer validation.

**Questions:**

N.A.

---

### Official Review · Reviewer_2wQg · 2025-10-31

**Soundness:** 1
**Presentation:** 1
**Contribution:** 1
**Rating:** 0
**Confidence:** 5

**Summary:**

This paper investigates the internal mechanisms of Llama 3.1 8B and 70B models on multi-argument addition tasks, introducing the concept of an "addition circuit." The authors identify specific attention heads and propose "state vectors" to capture how each argument is represented in the model’s activation space. Through interventions in middle layers, they confirm the causal role of these state vectors and show that they generalize across prompt formats, languages, and number representations (e.g., Arabic digits vs. word numerals).

**Strengths:**

1. this paper discuss the mechanism of multi-argument addition tasks.
2. this paper find some important attention heads and introduce "state vectors" to represent summands.

**Weaknesses:**

1. Limited Novelty in Task and Setting:
The core task—multi-argument single-token addition—is a straightforward extension of prior work, which already analyzed two-argument addition circuits in Llama 2 and Mistral. The paper acknowledges that carry propagation (the main complexity in multi-digit addition) is not addressed, and the multi-argument setting essentially reduces to independent single-digit additions, offering little conceptual advancement.

2. Methodologically incremental:
The method is not novel, which are closely follow established techniques in mechanistic interpretability, such as activation patching and function vectors. Only an application of existing tools to a slightly broader arithmetic setting.

3. Predictable and Repetitive Findings
The key results—specialized attention heads for operand positions, generalization across prompts/languages, and causal efficacy of interventions, and generalization for different forms—mirror conclusions already demonstrated in earlier studies on modular addition (Nanda et al., 2023) and arithmetic reasoning (Zhou et al., 2024; Zhang et al., 2024).

4. Poor Scholarly Presentation:
The writing reads more like a technical report than a scientific paper.

**Questions:**

see weakness

---

### Official Review · Reviewer_HjsP · 2025-10-31

**Soundness:** 1
**Presentation:** 2
**Contribution:** 1
**Rating:** 2
**Confidence:** 4

**Summary:**

The paper claims to uncover an addition circuit in Llama-3.1 8B and 70B: a sparse set of mid-layer attention heads that track each addend’s position for multi-argument integer addition. The authors define state vectors by averaging head-restricted contributions from source tokens to the final position, then perform causal interventions that subtract a state vector for value $c$ and add one for $c'$ to steer the predicted sum. They report that a few heads in specific layers are sufficient to change outputs in the expected way, and that the same state vectors transfer across two prompt templates as well as six languages with number words. They also present a brief observation for two multi-token numbers. The scope is restricted to numbers in [0,99] so that inputs and outputs are single tokens.

**Strengths:**

The setup is simple, reproducible, and clearly defined. The state-vector construction is explicit and easy to implement. The paper surfaces specific mid-layers and head sets where interventions have the largest effect, which may be a useful pointer for follow-up work. Cross-template and cross-language demonstrations are tidy, even if narrow.

**Weaknesses:**

1. Limited novelty and technical depth. The work repackages known activation-level techniques with average-based “state vectors,” without weight-level analysis, path-patching comparisons, or causal scrubbing.

2. Single-token simplification. All addends and outputs are forced into one token; this removes carry propagation complexity and makes linear steering at the final position unusually effective. The multi-token section is anecdotal.

3. Weak causality tests. Heads are selected by highest attention to operands, then used to build state vectors. There is no necessity test that keeps only those heads active or ablates them while measuring accuracy, nor controls with random heads or random vectors.

4. Intervention scope is narrow. Interventions are applied at a single layer and location, on curated prompts. No robustness to template noise, distractors, or perturbations of unrelated tokens is reported.

5. Claims of "semantic" state vectors are overstated. Transfer to a few languages does not establish semantic abstraction, especially given heavy constraints on number range and templating.

6. Generalization not demonstrated. The paper studies only two Llama models and acknowledges limited scope.

**Questions:**

1. Can you demonstrate necessity by ablating only the proposed head sets and showing that accuracy drops to chance while other heads are intact, and sufficiency by zeroing all other heads in those layers and recovering near-baseline accuracy. How sensitive are results to the chosen k heads.

2. How do your state-vector interventions compare to path patching or function-vector steering on the same prompts. Please include quantitative side-by-side results.

3. Do results hold when addends and outputs are multi-token with realistic carries, for example values in [0,9999] and sums that require two or three tokens. The current multi-token figure is anecdotal and lacks metrics. How do you make the judgements, with your eyes?

4. What happens if you shuffle positions or insert distractor numbers and nouns to test whether heads truly track semantic roles rather than fixed positional patterns.

5. Can you report controls: random-head selection, random state vectors with matched norm, and layer-misaligned vectors. This would calibrate the effect size of your proposed mechanism.

6. Beyond attention scores, did you examine value-vector content $W_V$ and output projections $W_O$ of the identified heads to argue mechanism rather than correlation.

---

### Meta-Review · Area_Chair_5KqV · 2026-01-11

**Summary:**

This paper presents a well-executed investigation into the internal mechanisms Llama-3.1 models use to perform multi-argument integer addition. By identifying a sparse set of mid-layer attention heads that appear to track individual addends—regardless of position—and introducing “state vectors” to formalize this representation, the authors extend prior work on arithmetic in LLMs in a meaningful direction. The demonstration of cross-template and cross-lingual transfer for these state vectors is particularly interesting and suggests a degree of abstraction in how the model encodes numerical information. However, the paper is weak in technical novelty and presentation.

**Reviewer Scores:**

NA

---

### Decision · Program_Chairs · 2026-01-26

Reject